# Synergistic Activity and Mechanism of Sanguinarine with Polymyxin B against Gram-Negative Bacterial Infections

**DOI:** 10.3390/pharmaceutics16010070

**Published:** 2024-01-03

**Authors:** Luyao Qiao, Yu Zhang, Ying Chen, Xiangyin Chi, Jinwen Ding, Hongjuan Zhang, Yanxing Han, Bo Zhang, Jiandong Jiang, Yuan Lin

**Affiliations:** 1State Key Laboratory of Bioactive Substance and Function of Natural Medicines, Institute of Materia Medica, Chinese Academy of Medical Sciences and Peking Union Medical College, Beijing 100050, China; qiaoly_pumc@163.com (L.Q.); zhangyu@imm.ac.cn (Y.Z.); chenyinga@imm.ac.cn (Y.C.); chixiangyin@imm.ac.cn (X.C.); dingjinwen@imm.ac.cn (J.D.); m15611585399@163.com (H.Z.); hanyanxing@imm.ac.cn (Y.H.); 2Department of Pharmacy & State Key Laboratory of Complex Severe and Rare Diseases, Peking Union Medical College Hospital, Peking Union Medical College, Chinese Academy of Medical Sciences, Beijing 100730, China; zhangbopumch@163.com

**Keywords:** Gram-negative bacteria, polymyxin B, sanguinarine, combinations, bacterial membrane

## Abstract

Compounds that potentiate the activity of clinically available antibiotics provide a complementary solution, except for developing novel antibiotics for the rapid emergence of multidrug-resistant Gram-negative bacteria (GNB). We sought to identify compounds potentiating polymyxin B (PMB), a traditional drug that has been revived as the last line for treating life-threatening GNB infections, thus reducing its nephrotoxicity and heterogeneous resistance in clinical use. In this study, we found a natural product, sanguinarine (SA), which potentiated the efficacy of PMB against GNB infections. The synergistic effect of SA with PMB was evaluated using a checkerboard assay and time–kill curves in vivo and the murine peritonitis model induced by *Escherichia coli* in female CD-1 mice in vivo. SA assisted PMB in accelerating the reduction in bacterial loads both in vitro and in vivo, improving the inflammatory responses and survival rate of infected animals. The subsequent detection of the intracellular ATP levels, membrane potential, and membrane integrity indicated that SA enhanced the bacterial-membrane-breaking capacity of PMB. A metabolomic analysis showed that the inhibition of energy metabolism, interference with nucleic acid biosynthesis, and the blocking of L-Ara4N-related PMB resistance may also contribute to the synergistic effect. This study is the first to reveal the synergistic activity and mechanism of SA with PMB, which highlights further insights into anti-GNB drug development.

## 1. Introduction

Gram-negative bacteria (GNB) infections are one of the most significant public health problems in the world due to their high resistance to antibiotics [1,2]. These bacteria play significant clinical roles in hospitals because they put intensive care unit (ICU) patients at high risk and lead to high morbidity and mortality rates. World Health Organization (WHO) reports have consistently emphasized the great need for antibiotics to treat GNB infections [3]. Epidemiological studies have shown that the GNBs with the highest infection rates worldwide are *Klebsiella pneumoniae*, *Escherichia coli*, *Pseudomonas aeruginosa*, and *Acinetobacter baumannii*, which are also the highest-priority pathogens in the WHO list for novel antibiotic development [3,4]. All of these factors indicate the great treatment challenge for clinical GNB infections.

The highly asymmetrical structure of the outer membrane (OM) serves as a unique permeability barrier that protects GNBs from toxic environmental factors [5]. Lipo-polysaccharide (LPS), arranged in the outer leaflet of the OM, is thought to act as the key permeability barrier, making the OM relatively impermeable to toxic compounds, including antibiotics and antimicrobial peptides [6]. In addition, the rapid emergence of multidrug-resistant GNBs (MDR-GNBs) is another key threat limiting the use of antibiotics [7]. Therefore, it is challenging to develop innovative compounds against GNB infections. Since the release of the WHO priority list in 2017, only five newly developed antibiotics against MDR-GNBs have been approved, none of which has a new mode of action, and all followed a fast-track process pathway [2]. In addition, the rapid development of drug resistance may render most available anti-GNB antibiotics (including polymyxins, tigecycline, minocycline, and sulbactam) useless [8,9]. Thus, new treatment strategies for GNB infections are urgently needed. It has been reported that when used in combination, these antibiotics are more effective against MDR-GNBs and significantly reduce the emergence of drug resistance [10]. In fact, drug combinations have become crucial regimens for clinical treatment, which have also been proposed as promising therapeutic strategies to overcome drug resistance and improve the efficacy of monotherapy regimens in GNB infections.

Polymyxins, the lipopeptide antibiotics, have been shown to be potent against MDR-GNBs, particularly *K. pneumoniae*, *E. coli*, *A. baumannii*, and *P. aeruginosa* [11]. Previous studies have confirmed that polymyxins mainly act on the LPS of GNBs to damage the integrity of the OM, leading to cell dysfunction until death [11]. The toxicity (mainly neurotoxicity and nephrotoxicity) and high heterogeneous resistance of polymyxins once limited its clinical use [12]. However, polymyxins have been revived as the last-line therapeutic option for treating life-threatening infections caused by MDR-GNBs [13]. In clinics, polymyxin-based combination therapy has been widely used to enhance the clinical effect and decrease the incidence of heterogeneous resistance [2,14]. The antibiotics commonly used clinically in combination with polymyxins are carbapenems, sulbactam, and tigecycline [15,16,17]. However, these combinations are still limited in efficacy and require high doses of polymyxins, which can easily cause polymyxin-related adverse drug reactions [18,19]. Colistin (polymyxin E) and PMB (polymyxin B) are two currently available polymyxins. Compared with colistin, PMB has favorable clinical pharmacologic properties and is widely adopted [14]. Therefore, the search for compounds that potentiate the antibacterial effect and lower the toxic dosage of PMB is an urgent need for the clinical treatment of MDR-GNB infections.

Sanguinarine (SA) is a benzophenanthridine alkaloid extracted from *Sanguinaria canadensis* and shows anti-GNB activity through various mechanisms. SA restrains the proliferation of GNBs by blocking cytokinesis by hindering FtsZ assembly and bundling or inhibiting *P. aeruginosa* 2-ketogluconate utilization by interfering with the function of KguD or KguK proteins [20,21]. In addition, SA effectively synergizes with aminoglycoside in killing GNBs due to the enhanced membrane permeability and increased intracellular ROS levels [22]. Thus, SA has the potential to synergize with other antibiotics to increase the antibacterial effect.

Here, we report a promising combination of SA with PMB for the treatment of GNB infections. We found that SA could promote the antibacterial activity of PMB in vitro and in an in vivo murine peritonitis model, probably by enhancing the breakage of GNB membranes. SA was shown to be effective in potentiating the activity of PMB against a variety of GNB strains for the first time, which also highlights its significant clinical potential for the treatment of GNB infections.

## 2. Materials and Methods

### 2.1. Reagents and Chemicals

Sanguinarine chloride was purchased from MedChamExpress (Monmouth Junction, NJ, USA), and polymyxin B was purchased from Inalco (Tuscany, Italy). The natural product library was obtained from MedChamExpress (Monmouth Junction, NJ, USA). Luria–Bertani (LB) medium and LB agar medium were purchased from BD Biosciences (Franklin Lakes, NJ, USA). Mouse IL-6/IL-1β/TNF-α ELISA kits for the detection of cytokines were purchased from Shanghai Enzyme-linked Biotechnology (Shanghai, China). ATP and AKP kits were purchased from Beyotime Biotechnology (Shanghai, China). BacLight Bacterial Membrane Potential Kits were obtained from Invitrogen (Carlsbad, CA, USA). TO-PRO-3 dye was purchased from AAT Bioquest (Pleasanton, CA, USA).

### 2.2. Bacterial Strains and Growth Conditions

A total of 7 bacterial strains employed in this study were purchased from BeNa Culture Collection (Beijing, China), including 3 *Escherichia coli* standard strains (ATCC 25922, 11775, 9637, Manassas, VA, USA), a resistant *Escherichia coli* strain (ATCC 35218, an *E. coli* strain producing extended-spectrum *β*-lactamases and resistant to *β*-lactam antibiotics), a *Klebsiella pneumoniae* strain (ATCC 700603, a *K. pneumoniae* strain producing extended-spectrum *β*-lactamases and resistant to *β*-lactam antibiotics), a *Pseudomonas aeruginosa* strain (ATCC 27853), and an *Acinetobacter baumannii* strain (ATCC 19606). All bacterial strains were cultured with LB medium and stored at −80 °C.

### 2.3. Minimal Inhibitory Concentration (MIC) Determination

The antibacterial activities of compounds were determined according to the 2021 Clinical and Laboratory Standards Institute drug sensitivity standard protocol [23]. Bacteria (1 × 10^6^ CFU/mL) were incubated with the compounds in a 96-well plate for 12 h at 37 °C. The compounds were in two-fold dilutions, with final concentrations of PMB ranging from 0.0625 to 8 μg/mL and those of SA ranging from 4 to 512 μg/mL. MICs were defined as the lowest concentrations of the compounds to cause no growth of bacteria.

### 2.4. Natural Product Library Screening

The natural product library, a commercial library containing 220 natural products, was used for the screen. In the initial screen, *E. coli* ATCC 25922 cells (1 × 10^6^ CFU/mL) were added to a 96-well plate, followed by the addition of PMB (1/2 × MIC) and compounds (10 μM). The microtiter plates were recorded with a plate reader at 600 nm after incubation at 37 °C for 12 h. The compounds in the wells with OD_600_ < 0.1 were regarded as the initial positive ones [22]. Subsequently, we performed a secondary screen. Different concentrations of the initial positive ones in 2-fold dilutions were further investigated by combining them with PMB (1/2 × MIC). Compounds that resulted in bacterial growth of OD_600_ < 0.1 with the lowest concentration were considered hits.

### 2.5. Checkerboard Assay

The synergistic effect of PMB and SA was determined by performing standard checkerboard broth microdilution assays. PMB and SA were serially diluted in eight-fold steps. OD_600_ was examined after incubation at 37 °C for 12 h. The FICI was calculated as below to analyze the synergistic effect using the concentration with the highest combination effects:$$\mathrm{FICI}=\frac{\mathrm{MIC}\;\mathrm{of}\;\mathrm{PMB}\;\mathrm{in}\;\mathrm{the}\;\mathrm{combination}}{\mathrm{MIC}\;\mathrm{of}\;\mathrm{PMB}\;\mathrm{alone}}+\frac{\mathrm{MIC}\;\mathrm{of}\;\mathrm{SA}\;\mathrm{in}\;\mathrm{the}\;\mathrm{combination}}{\mathrm{MIC}\;\mathrm{of}\;\mathrm{SA}\;\mathrm{alone}}$$

A synergistic effect is defined as FICI ≤ 0.5, while an antagonistic effect is defined as FICI ≥ 4.0. In addition, an indifferent effect is defined as 0.5 < FICI < 4.0 [24].

### 2.6. Time–Kill Curves

Time–kill curves were examined for PMB and SA alone or in combination at their FICI concentrations for different strains at 37 °C with an initial bacterial concentration of 1 × 10^6^ CFU/mL. A 200 μL aliquot at different time points (0, 2, 4, 6, 8, 12, 24 h) was obtained and plated on LB agar plates after serial dilution. Bacterial colonies were counted as the log of viable cells (CFU/mL) after incubation.

### 2.7. E. coli-Induced Peritonitis Model

Female CD-1 mice (18–20 g) were purchased from Vital River Laboratories (Beijing, China). All animals were housed under controlled humidity (30–70%) and temperature (22 ± 3 °C) and a 12 h light–dark cycle. The mice were then fasted for 12 h before bacteria were administered. 

Peritonitis was induced by infecting the mice intraperitoneally with 3–4 × 10^4^ CFU of *E. coli* ATCC 25922 in 5% mucin. There were five experimental groups: (1) vehicle (treated with sterile saline); (2) MO (infected and treated with sterile saline); (3) PMB (infected and treated with 0.5 mg/kg PMB); (4) SA (infected and treated with 10 mg/kg SA); (5) COM (infected and treated with 0.5 mg/kg PMB and 10 mg/kg SA). PMB was administered subcutaneously, and SA was administered intragastrically at 2 and 6 h post-infection. The deaths of the mice were recorded for 7 days after infection to calculate the survival rates (18 mice/group). The livers, kidneys, and spleens were aseptically removed for histopathology at 12 h post-infection. The tissues were immersed in 4% paraformaldehyde. After paraffin embedding, 5 μm sections were cut and stained with hematoxylin and eosin (H&E) to assess the pathological changes using light microscopy. In addition, body fluids (blood and peritoneal irrigation fluid) and tissues (livers, kidneys, and spleens) were collected to measure bacterial burden (5 mice/group) at 12 h post-infection. For body fluids, samples were diluted and plated on LB agar plates at 37 °C for 12 h for CFU counting. For tissues, samples were homogenized and diluted in sterile saline before counting.

Complete blood count, ELISA, and quantitative reverse transcription PCR (RT-qPCR) were used to analyze the inflammatory response of blood and tissues at 12 h post-infection. Details of the primers used for RT-qPCR are provided in Appendix A. The relative mRNA levels were calculated based on the 2^−ΔΔCt^ method according to Livak and Schmittgen [25]. The complete blood count was performed with a Nihon Kohden MEK-7222K Hematology Analyzer (Nihon Kohden, Rosbach vor der Höhe, Germany) for leukocyte and neutrophil counts. The cytokines IL-6, IL-1*β*, and TNF-*α* in plasma were detected by ELISA kits. The gene expression levels of IL-6, IL-1*β*, and TNF-*α* in tissues were determined by RT-qPCR using Applied Biosystem ABI 7500 Fast (ThermoFisher Scientific, Waltham, MA, USA).

### 2.8. Intracellular ATP Level Detection

*E. coli* ATCC 25922 (1 × 10^6^ CFU/mL) was incubated at 37 °C and treated with the compounds for 2 h in the following groups: (1) Ctrl (without compounds); (2) PMB (1/16 × MIC, 0.03 μg/mL); (3) SA (1/4 × MIC, 4 μg/mL); (4) COM (1/16 × MIC PMB in combination with 1/4 × MIC SA). The cultures were sedimented and lysed with a tissue grinding machine (Servicebio, Wuhan, China). Then, the supernatants were applied to detect the intracellular ATP concentration with an ATP assay kit using a luciferin–luciferase fluorescence assay. The ATP in the samples provides the energy to enable firefly luciferase to catalyze luciferin to produce fluorescence that is proportional to the ATP concentration. The luminescence intensity was measured using an EnSpire Multimode Reader (PerkinElmer, Waltham, MA, USA). An ATP standard curve was plotted by measuring different concentrations of ATP. Three biological replicates with at least three technical replicates were prepared for each group.

### 2.9. Determination of Membrane Potential

The bacterial membrane potential was determined using a FACSCalibur flow cytometer from BD Biosciences (Franklin Lakes, NJ, USA) using a BacLight Bacterial Membrane Potential Kit. *E. coli* ATCC 25922 (1 × 10^6^ CFU/mL) was incubated at 37 °C for 2 h and treated with the compounds as follows: (1) Ctrl (without compounds); (2) CCCP (carbonyl cyanide 3-chlorophenylhydrazone, 5 μM, as a fully depolarized control); (3) PMB (1/16 × MIC, 0.03 μg/mL); (4) SA (1/4 × MIC, 4 μg/mL); (5) COM (1/16 × MIC PMB in combination with 1/4 × MIC SA). The fluorescent membrane potential indicator dye 3,3-diethyloxacarbocyanine iodide (DiOC_2_), which exhibits red fluorescence at a high membrane potential and green fluorescence at a low membrane potential, was added to each culture for 30 min. CCCP, a proton ionophore that can destroy the membrane potential by eliminating the proton gradient, was used as a fully depolarized control. Additionally, 1 mM EDTA was added to improve the absorption of CCCP [26]. The value of the red/green fluorescence ratio was calculated to evaluate the membrane damage. Three biological replicates were prepared for each group.

### 2.10. Determination of Membrane Integrity

Bacterial membrane integrity was determined with a FACSCalibur flow cytometer from BD Biosciences (Franklin Lakes, NJ, USA) using the fluorescent dye TO-PRO-3. TO-PRO-3 is able to permeate into membrane-damaged bacteria and fluorescently label nucleic acids. The experimental groups were as follows: (1) Ctrl-v (viable control); (2) Ctrl-d (dead control); (3) PMB (1/16 × MIC, 0.03 μg/mL); (4) SA (1/4 × MIC, 4 μg/mL); (5) COM (1/16 × MIC PMB in combination with 1/4 × MIC SA). Fresh colonies were grown overnight in LB medium (one colony per 10 mL of LB) as the initial bacterial solution. Each group of bacteria was cultured at 37 °C for 2 h. The dead control was then heat-killed at 72 °C for 30 min [27]. Cells were washed with PBS and stained with 100 nM TO-PRO-3 for 30 min at room temperature. Stained cells were analyzed on the flow cytometer. TO-PRO-3 was detected through a 695 nm long-pass emission filter. The median fluorescence intensity (MFI) ratio, which is defined as the MFI of each group over the MFI of the Ctrl-v group, was used to measure the degree of membrane damage. Three biological replicates were prepared for each group.

### 2.11. Electron Microscopy

*E. coli* ATCC 25922 was collected for scanning electron microscopy (SEM) after being treated with the compounds for 2 h. PMB (1/16 × MIC, 0.03 μg/mL), SA (1/4 × MIC, 4 μg/mL), or their combination was used. Bacterial pellets were resuspended and fixed in 2.5% glutaraldehyde at 4 °C for 12 h, followed by PBS washing. The pellets were then harvested and gradually dehydrated with various concentrations of ethanol (30%, 50%, 70%, 80%, 90%, and 100%) for 10 min each. Finally, the samples were vacuum-sputter-coated with gold and examined using SEM.

For transmission electron microscopy (TEM), bacterial pellets were fixed with 1% osmic acid and dehydrated with acetone at different concentrations in sequence (50%, 70%, 80%, and 90% for 15 min each and 100% for 20 min). Samples were infiltrated with resin and stained for TEM observation.

### 2.12. Metabolomic Analysis

Global metabolic variations in *E*. *coli* ATCC 25922 were analyzed by using untargeted metabolomics. *E. coli* ATCC 25922 was cultured to the logarithmic phase with 1 × 10^8^ CFU/mL. Four groups were examined as follows: (1) Ctrl (without compounds); (2) PMB (2 × MIC, 1 μg/mL); (3) SA (2 × MIC, 32 μg/mL); (4) COM (PMB-SA combination group with the same concentrations). Six biological replicates were prepared for each group. All of the samples were collected at 1 h and normalized by optical density (OD_600_). The cells were washed and resuspended in methanol/water (4:1, *v/v*). After adding chloroform, the samples were sonicated. Sequentially, as a generic internal standard, L-2-chlorophenylalanine (0.3 mg/mL) was added. Then, samples were ultrasonically extracted for 20 min and centrifuged (13,000 rpm, 4 °C) for 10 min. Supernatants were stored in vials for HPLC-MS/MS analysis.

Samples were analyzed on a Q-Exactive plus an Orbitrap MS (Thermo Fisher Scientific, Waltham, MA, USA) equipped with a heated electrospray ionization (ESI) source (Thermo Fisher Scientific, Waltham, MA, USA), which was operated in both positive and negative ESI ion modes with a detection range of 70 to 1000 m/z. The MS was coupled to an ACQUITY UPLC I-Class plus (Waters Corporation, Milford, MA, USA) with a ZIC-pHILIC column (1.8 μm, polymeric, 100 × 2.1 mm; ACQUITY UPLC HSS T3, Waters Corporation, Milford, MA, USA). Water and acetonitrile, both containing 0.1% formic acid, were used as mobile phases A and B, respectively. The flow rate was 0.35 mL/min, and the injection volume was 2 μL.

### 2.13. Statistical Analysis

The data were calculated with GraphPad Prism 8.0 software (San Diego, CA, USA). The survival rate was analyzed using a log-rank (Mantel–Cox) test. Differences between group were calculated using Welch’s one-way ANOVA with Dunnett’s T3 multiple-comparison test and Student’s *t*-test. All data are expressed as mean ± SD (error bars). A *p*-value of <0.05 was defined as statistically significant (* *p* < 0.05, ** *p* < 0.01, *** *p* < 0.001, **** *p* < 0.0001).

## 3. Results

### 3.1. Screening for Compounds Potentiating the Activity of PMB against E. coli

To investigate compounds that could potentiate the antibacterial effect of PMB against GNBs, the *E. coli* standard strain ATCC 25922 was used. We screened compounds from a natural product library of 220 compounds. First, the minimal inhibitory concentration (MIC) of PMB against *E. coli* ATCC 25922 was determined as 0.5 μg/mL. In the initial screen, PMB (1/2 × MIC, 0.25 μg/mL) combined with 10 μM compounds were added to the plate. Using OD_600_ = 0.1 as the cut-off point, compounds with OD_600_ < 0.1 were considered to promote the antibacterial activity of PMB. The results of the initial screen showed that four compounds potentiated the activity of PMB (Figure 1a). The positive rate was 1.8%. Then, in the secondary screen, the concentration of PMB was kept at 1/2 × MIC. Meanwhile, the concentrations of the four initial positive compounds were changed to 2-fold gradient dilutions (0.06–10 μM) to explore the compound that could inhibit bacterial growth (OD_600_ < 0.1) at the lowest concentration when combined with 1/2 × MIC PMB. It was shown that sanguinarine (SA) exhibited potentiating activity with 1/2 × MIC PMB at a concentration of 1.25 μM, lower than those of the other three compounds (Figure 1b). Thus, we selected SA for further in vitro and in vivo antibacterial activity analyses combined with PMB. The structure of SA is shown in Figure 1c.

### 3.2. In Vitro and In Vivo Antibacterial Activity Assays

#### 3.2.1. In Vitro Antibacterial Activity of PMB-SA Combinations

To evaluate the in vitro activities of the compounds against GNBs, different GNB strains (*E. coli* ATCC 25922, *E. coli* ATCC 11775, *E. coli* ATCC 9637, *E. coli* ATCC 35218, *K. pneumoniae* ATCC 700603, *P. aeruginosa* ATCC 27853, and *A. baumannii* ATCC 19606) were tested. The MICs of PMB and SA alone against these strains are summarized in Table 1. PMB showed an inhibitory effect on all of the tested GNB strains, with MICs of 0.5–2 μg/mL. Compared with PMB, SA showed moderate activity against the tested strains, with MICs of 16–256 μg/mL.

A checkerboard analysis was used to determine the potency of the antibiotic combination compared with their individual activities. The results are summarized in Table 1. The fractional inhibitory concentration index (FICI) was used to reflect the combined antimicrobial effect. When combined with sub-MIC concentrations of SA, the MICs of PMB against GNB strains showed a dose-dependent decrease from 0.5–2 μg/mL to 0.125–0.25 μg/mL (Table 1, Figure 2a). FICIs were calculated using the combination concentrations with the highest synergistic activity, which was 1/8–1/4 × MIC of PMB in combination with 1/8–1/4 × MIC of SA. Overall, all of these results indicated that PMB-SA combinations presented remarkably synergistic activities against the tested GNB strains in vitro.

Time–kill assays were performed to investigate the synergistic effect of PMB and SA against all tested strains over time. The concentrations of PMB and SA at FICIs were used to construct the time–kill curves. As shown in Figure 2b, PMB alone (1/8–1/4 × MIC) induced bacterial death in 2–4 h, with the bacteria growing steadily after 4 h, reaching roughly 10^8^–10^10^ CFU/mL at 12–24 h. SA alone displayed no reduction in CFU counts. In contrast, PMB combined with SA caused remarkable growth inhibition against all seven strains over the 0–24 h period, although the effect on *P. aeruginosa* ATCC 27853 was reduced after 24 h. Thus, the combination of PMB and SA showed a significant synergistic effect on all tested strains, exhibiting a typical bactericidal mode.

#### 3.2.2. In Vivo Antibacterial Activity of PMB-SA Combination in *E. coli*-Induced Peritonitis Model

To investigate the antibacterial effect of the PMB-SA combination in vivo, we adopted a peritonitis model induced by *E. coli* infection, which has been widely used as an experimental model for sepsis [28,29,30]. The scheme of the experimental protocol is shown in Figure 3a. The survival rate for each group was calculated to evaluate the curative effect of PMB, SA, and their combination. The results showed that all mice from the model group without treatment died within 24 h (Figure 3b). By day 7, SA alone had shown no influence on the survival rate. When PMB was administered alone, the survival rate was improved to 33.3%. Interestingly, the combination of PMB and SA significantly increased the survival rate to 77.8% (*p* < 0.01), more than 2.3 times that observed with PMB alone. The results indicated that SA showed an obvious synergistic effect with PMB on the survival rates of mice with *E. coli-*induced peritonitis.

We also conducted a histological analysis of the livers, kidneys, and spleens in each group after *E. coli* infection by using H&E staining (Figure 3c). Compared to the vehicle group, significant pathological changes in the model group appeared. In the livers and kidneys, the overall arrangement and shape of the cells were abnormal, with a decreased volume of the cell nucleus, increased intercellular space, and disordered cell structure. The renal tissues exhibited severe vacuolar degeneration of renal tubular epithelial cells and inflammatory cell infiltration. In spleens, the splenic trabeculae were broken, and the white pulp areas in the model group were increased, which is reflected by the statistical analysis in Appendix A. Such changes were reversed after PMB, SA, or PMB-SA treatments, especially in the PMB-SA combination treatment group.

Bacterial loads in body fluids (blood and peritoneal lavage fluids) and tissues (livers, kidneys, and spleens) of each group were examined next. As shown in Figure 3d, colony counts in model groups were significantly increased due to the infection (10^6^–10^9^ CFU/mL in body fluids, 10^6^–10^8^ CFU/mg in tissues). SA alone had almost no antibacterial effect, whereas PMB alone led to an approximately 10^2^-fold (both in body fluid and tissues) decrease in bacterial loads compared to the model groups. However, the bacterial loads were 10^2–^10^3^-fold (10^3^ in body fluids, 10^2^ in tissues) lower when treated with the PMB-SA combination than with PMB alone, indicating that the PMB-SA combination exhibited synergistically effective bacterial-killing activity.

To better understand the effect of the PMB-SA combination on *E. coli-*induced peritonitis, inflammatory cells and cytokines were measured. The results showed that intraperitoneal injection of *E. coli* caused a severe inflammatory response by greatly increasing leukocytes and neutrophils in the blood (Appendix A), as well as the expression of cytokines like IL-6, IL-1*β*, and TNF-*α* in the plasma (Figure 3d) and tissues (Appendix A). The inflammatory cells and cytokines were slightly decreased in both, the PMB- and SA-alone groups. In contrast, there was a remarkable decrease in leukocytes, neutrophils, and the expression of IL-6, IL-1*β,* and TNF-*α* in PMB-SA combination groups, indicating the attenuation of the inflammatory response in *E. coli-*induced peritonitis after PMB-SA combination therapy. Overall, these results strongly demonstrated that PMB and SA had synergistic in vivo therapeutic effects on *E. coli-*induced peritonitis.

### 3.3. Exploring the Synergic Mechanism of SA with PMB

#### 3.3.1. Determination of Intracellular ATP

The effect of the compounds on the integrity of the bacterial membrane was measured by determining the intracellular ATP content. When the bacterial membrane is disrupted, the permeability of the cell increases, leading to decreased intracellular ATP. Bacterial cells were lysed, and the supernatants were applied for ATP detection using a luminescence-based assay. According to Figure 4a, intracellular ATP in both PMB and SA groups decreased. Importantly, the intracellular ATP decreased significantly in the PMB-SA combination group compared to PMB alone (1/16 × MIC PMB) (*p* < 0.001), indicating that the combination of PMB and SA exhibited significant synergistic damage to the bacteria membrane.

#### 3.3.2. Fluctuation of Bacterial Membrane Potential

To further investigate the damage from the compounds to bacterial cell membranes, membrane potential assays were conducted using a fluorescence-based kit. The membrane potential decreased when it was disrupted by the neutralization of charges across the membrane. DiOC_2_, a fluorescent probe that shifts from red to green when the membrane potential decreases, was used. After adding DiOC_2_, each group was measured with a flow cytometer. CCCP, a proton ionophore, was used as the positive control for the depolarization of the membrane. As shown in Figure 4b, we found that the membrane potential was decreased in the presence of PMB or SA alone. However, a dramatic decrease in membrane potential was observed in the PMB and SA combination group compared with PMB alone (1/16 × MIC) (*p* < 0.05). This implied that SA synergistically enhanced the membrane damage caused by PMB.

#### 3.3.3. Changes in Bacterial Membrane Permeability

Membrane breakage caused by compounds leads to DNA exposure. The membrane-impermeable dye TO-PRO-3, which exhibits fluorescence when it binds to double-stranded DNA in bacteria, was detected with a flow cytometer to evaluate the membrane-damage activity of the compounds. As shown in Figure 4c, the fluorescence intensity of the dead cell control was significantly shifted upward compared to the viable cell control. Cells exposed to PMB alone contained both dead and viable cells (44.7% dead cells, 55.2% viable cells), and the SA group was mostly viable (10.4% dead cells, 89.6% viable cells). Compared with PMB alone, the percentage of dead cells in the PMB-SA combination group increased from 44.7% to 59.6%. The membrane damage can be reflected by MFI, the ratio of which was used to show the degree of TO-PRO-3 binding in each group. It was shown that the MFI ratio of the PMB-SA group was significantly higher than that of PMB alone (1/16 × MIC) (*p* < 0.05), suggesting that SA synergistically accelerated the damaging effect of PMB on the bacterial membrane, consistent with the previous results.

#### 3.3.4. Observation of Cell Morphology Using Electron Microscopy

The surface changes in *E. coli* were observed using SEM. As shown in Figure 4d, bacteria treated with either PMB or SA alone at the indicated concentrations generated mild damage to the cell membrane with little protrusions. In contrast, we found more damaged membranes with protrusions in the PMB-SA-treated group, indicating the promotion of the leakage of cell contents by SA with PMB. The results showed a synergistic effect of SA with PMB on membrane breakage.

TEM was also used for further observation of the ultrastructural changes in *E. coli*. The PMB (1/16 × MIC)-treated group exhibited some tubular appendages, mesosomes, and inhomogeneous cytoplasm with an abnormally increasing electron density. Compared to the PMB group, more tubular appendages, mesosomes, and residual cell membrane structures were observed when treated with the PMB-SA combination. The TEM results also showed that the PMB-SA combination had strong synergistic effects on bacterial membrane breakage.

#### 3.3.5. Metabolomic Analysis

To explore the metabolic mechanisms by which SA potentiated the anti-GNB effect of PMB, HPLC-MS-based metabolomics was used to compare the metabolic changes in *E. coli* ATCC 25922 in the presence of PMB, SA, and both compounds. The relative standard deviations of all samples were below 30%, which is acceptable for metabolomic studies [31]. There were 3434 putative metabolites identified in an untargeted metabolomic study, including lipids, nucleotides, amino acids, and carbohydrates. Principal component analysis (PCA) results showed that the control group and treated groups (PMB, SA, and combination) were significantly separated (Appendix A). Compared with the control group, a total of 489 metabolites were determined, where 214, 244, and 264 differentially abundant metabolites were found in the PMB, SA, and combination groups, respectively. Among these differentially abundant metabolites, 80 existed only in the combination group (Appendix A).

Our previous studies showed that SA potentiated the membrane-breaking effect of PMB. Thus, we focused on the detection of lipid metabolites and LPS-related metabolic pathways associated with membrane perturbation. As shown in Appendix A, the most significantly changed metabolites were lipid metabolites, which mainly contained fatty acids (FAs) and glycerophospholipids (GLPs), the essential components of the bacterial cell membrane. Therefore, we examined the changes in FAs and GLPs in different groups (Figure 5a). Most of the FAs showed a moderate reducing trend in the PMB group, while they were significantly decreased in the PMB-SA combination group. In addition, compared to the untreated group, GLPs increased in the PMB-SA combination group, while they were unchanged in the PMB group. The results showed that the PMB-SA combination group caused a more drastic perturbation of lipid metabolites in the bacterial membrane than PMB alone, which indicated the synergistic antibacterial activity of SA with PMB.

Metabolic pathways are a series of chemical reactions resulting in the anabolism or breakdown of metabolites in bacterial cells. LPS is not only a key component of the bacterial membrane but also the target of PMB. Thus, the changes in metabolites in LPS-related metabolic pathways are vital to understanding the synergistic mechanism of PMB and SA. As shown in Figure 5b, the PMB-SA combination treatment induced significant variations in intermediate metabolites involved in amino sugar/nucleotide sugar metabolism, glycolysis, and pentose phosphate pathways, which led to changes in the biosynthesis of LPS, while the PMB group remained almost the same. D-Fructose-2,6-biphosphate (log_2_FC = 1.84), D-Fructose (log_2_FC = 2.45), D-Glucose (log_2_FC = 1.94), and 2-Oxoglutarate (log_2_FC = 2.16) in glycolysis and N-acetyl-D-glucosamine (log_2_FC = 0.87) and D-Glucosamine (log_2_FC = 0.81) in amino sugar/nucleotide sugar metabolism were significantly increased after treatment with the PMB-SA combination compared to PMB alone (*p* < 0.01). Arginine metabolism, a pathway strongly interrelated with glycolysis, also significantly increased in the PMB-SA combination group compared to the PMB group (*p* < 0.01). Thus, the metabolites downstream of the LPS biosynthesis pathways increased more in the PMB-SA combination group than in the group treated with PMB alone, which showed more pronounced perturbations of LPS biosynthesis in the combination group. In addition, resistance pathways associated with LPS modifications, the dominant resistance mechanism for PMB in GNBs, were detected [32,33]. *α*-D-Glucose (log_2_FC = −0.72), *α*-D-galactose (log_2_FC = −0.37), UDP-D-galactose (log_2_FC = −2.29), and UDP-L-Ara4FN (log_2_FC = −0.58), the key metabolites of L-Ara4N biosynthesis pathways associated with LPS modification, were decreased in the PMB-SA combination group but increased in the PMB-treated group (Figure 5c). These results suggested that the PMB-SA combination caused more significant perturbations of LPS and resistance inhibition by PMB through L-Ara4N biosynthesis pathways, increasing the susceptibility of bacteria to PMB.

Moreover, we focused on the metabolic changes in energy metabolism and DNA and RNA biosynthesis to explore other mechanisms of enhanced antibacterial effects after combined PMB-SA use [34,35]. First, oxidative phosphorylation, pantothenate, and coenzyme A biosynthesis pathways, essential for energy metabolism, were detected (Appendix A). The metabolites of nicotinate and nicotinamide metabolism in oxidative phosphorylation were significantly reduced in the PMB-SA combination group compared with PMB alone (*p* < 0.01), such as nicotinamide adenine dinucleotide (log_2_FC = −4.80) and niacinamide (log_2_FC = −3.50). In addition, pyrophosphate (log_2_FC = −1.96) and phosphate (log_2_FC = −1.28) were decreased after PMB-SA combination treatment, compared to the increased levels in PMB groups. The reduction in phosphate caused a disturbance in the QseE/QseF regulatory system, a quorum-sensing system related to membrane stability, which also supports previous results showing synergistic membrane damage by the PMB-SA combination [36]. The levels of succinate (log_2_FC = −1.15) and citrate (log_2_FC = −1.93) in the tricarboxylic acid cycle, associated with pantothenate and coenzyme A biosynthesis, were reduced in the PMB-SA combination group, while they were largely increased in the PMB group. In addition to energy metabolism, pantothenate and coenzyme A biosynthesis pathways also provided substrates for FA metabolites and LPS biosynthesis [37,38]. Thus, the decrease in metabolites in these pathways further confirmed the synergistic effect of SA with PMB in FA and LPS biosynthesis pathways. Second, in DNA and RNA biosynthesis pathways, the combination group showed a markable reduction in hypoxanthine, adenine, and uracil metabolism and a partial increase in guanine, thymine, and cytosine compared to PMB alone (Appendix A). N2, N2-Dimethylguanosine (log_2_FC = −2.09), closely related to RNA stability, was decreased in the combination group [39]. In summary, the PMB-SA combination induced decreased energy metabolism and DNA and RNA biosynthesis, thereby affecting bacterial survival, reproduction, and drug resistance, which might assist in promoting the synergistic activity of SA with PMB.

Taken together, our results show that the combination of PMB and SA induced dramatic perturbations of the bacterial membrane. In addition, the inhibition of energy metabolism, interference with DNA and RNA biosynthesis, and the blocking of L-Ara4N-related PMB resistance might also contribute to the synergistic mechanisms of SA with PMB.

## 4. Discussion

The rapid emergence of polymyxin-resistant strains has brought the utility of polymyxin monotherapy into question. One strategy for overcoming polymyxin-relevant drug resistance without augmenting toxicity is to decrease polymyxin exposure by combining it with other agents. In this study, we found a natural product, SA, which exhibited a synergistic antibacterial effect with PMB, mainly by enhancing bacterial membrane damage, providing a new insight into treating GNB infections.

Combining polymyxins with antibiotic or non-antibiotic adjuvants has emerged as a novel solution against MDR-GNBs. Antibiotics that have been extensively investigated in combination with polymyxins included carbapenems, tigecycline, rifampin, sulbactam, chloramphenicol, telavancin, fosfomycin, vancomycin, and minocycline [40]. However, certain combinations, like the ciprofloxacin–ceftazidime combination, may induce resistance in *Pseudomonas aeruginosa* due to overexpressed efflux pumps [41]. Such cases are relatively rare when antibiotics are combined with non-antibiotic adjuvants, which are defined as compounds that are capable of re-sensitizing bacteria to antibiotics with no in vitro antibacterial activity themselves [42]. The potentiation mechanisms of these non-antibiotic adjuvants include increasing membrane permeability, inhibiting enzymes involved in antibiotic inactivation, blocking efflux pumps, and promoting biofilm formation [4]. Robin et al. found that the clinically available non-antibiotic drugs citalopram, sertraline, and spironolactone each showed synergistic effects with PMB [43]. Moreover, the natural products curcumin and tetrandrine enhanced the antibacterial activity of polymyxins against polymyxin-resistant strains [44,45]. Cannabinoids isolated from *Cannabis sativa* exhibited synergistic activity with PMB in vitro [46]. In this study, our results indicate that SA may have an advantage in reducing the risk of bacterial resistance to PMB. Compared to other natural product adjuvants, the synergistic activity of PMB with SA both in vitro and in vivo was explored, with the defined mechanism that SA enhances the bacterial membrane damage caused by PMB. All of these results indicate an advantage of SA in optimizing PMB combination therapy.

SA is a natural product with a wide range of pharmacological activities in tumors, cardiovascular diseases, neurodegenerative disorders, and inflammatory and infectious diseases [47,48,49,50,51]. SA can inhibit the proliferation and metastasis of lung carcinoma by regulating exosomes and induce human melanoma caspase-dependent cell death by promoting ROS generation [47,52]. SA was also shown to interfere with the aggregation of 42-amino acid amyloid *β* peptide, which is the main component of extracellular amyloid deposits, demonstrating its potential in neurodegenerative diseases [48]. It was reported that SA could inhibit platelet activation to attenuate thrombus formation, which contributes to its activity in cardiovascular disease [49]. For anti-infection activity, SA was shown to inhibit *Staphylococcus aureus* and methicillin-resistant *Staphylococcus aureus* (MRSA) with MICs of 1.56–6.25 µg/mL by inducing the release of membrane-bound cell wall autolytic enzymes and disrupting the bacterial membrane [53,54]. Zhang et al. also reported that SA could inhibit the growth of *Salmonella enterica* and *Providencia rettgeri* by interfering with the integrity of bacterial membranes [50,55]. There have been some reports on SA combined with other antibiotics. Obiang-obounou et al. found that SA synergistically enhanced the antibacterial activity of multiple antibiotics (ampicillin, oxacillin, norfloxacin, ciprofloxacin, and vancomycin) against MRSA [56]. Otherwise, SA was proven to synergize with colistin against six highly colistin-resistant primary clinical isolates [57]. Since PMB is currently one of the most effective drugs available for severe GNB infections, SA, which promotes the activity of PMB, has clinical significance [2]. It has been reported that SA itself could reduce the inflammatory response of macrophages in LPS-induced peritonitis [58], which might also contribute to the synergistic activity of PMB-SA in the *E. coli*-induced peritonitis model.

However, cytotoxicity induced by the binding of SA to Na^+^-K^+^-ATPase has been reported to be an important factor limiting its clinical applications [59]. Interestingly, SA exhibited relatively low toxicity when given by oral administration or in ionic form [59,60]. In addition, Cao et al. found a promising anti-cancer lead compound with higher selective cytotoxicity through SA-based structural modification, indicating the potential use of SA in anti-infectious diseases through structural modification [61]. Among polymyxins, colistin and PMB are two currently available types, both of which are considered to cause nephrotoxicity. Recent studies have shown that the prevalence of nephrotoxicity was significantly higher with colistin than with PMB [62,63]. Due to the dose-limiting adverse effect of nephrotoxicity, reducing the daily doses of PMB leads to a decrease in the risk of nephrotoxicity, which will subsequently result in subtherapeutic antibiotic concentrations and diminish bacterial killing. However, our study showed that the dose of PMB was reduced when combined with SA without attenuated efficacy. Thus, combinations of SA with low-dose PMB might be promising in clinical use.

Metabolomics is a comprehensive analysis of metabolites and metabolic pathways in a cell, tissue, or organism and can be applied in screening, diagnosis, prognosis, and treatment follow-up [64]. It has been widely used in pharmaceuticals to explore the mechanisms and therapeutic targets of drugs [65]. For infectious diseases, changes in the resistance, virulence factors, and adaptation characteristics of strains after drug treatment can be determined by metabolomics [66]. To date, the mechanisms of multiple polymyxin combination regimens have been investigated by using metabolomics. For example, Zhao et al. used metabolomics to explore the synergistic mechanism of PMB with rifampicin against *A. baumannii* AB5075, an MDR clinical isolate. By comparing changes in metabolites and metabolic pathways, they found that after PMB disrupted the bacterial outer membrane, rifampicin entered the intracellular space to inhibit DNA and RNA synthesis, which led to the synergistic activity of PMB with rifampicin [31]. A variety of mechanisms are proposed to mediate PMB synergism, including increased membrane perturbations, LPS modifications, the use of efflux pumps, the formation of capsules, and the overexpression of the outer membrane protein OprH [32,67,68]. In our study, using metabolomic analysis, we found that SA enhanced the membrane perturbations caused by PMB, consistent with the pharmacological results of the synergistic activity of PMB-SA in membrane disruption. Meanwhile, after treatment with PMB-SA, changes in the metabolites of L-Ara4N biosynthesis pathways, which have been proven to be associated with LPS modifications, were found. All of these findings serve as hints for following studies on the significance of PMB-SA synergism. In order to further explore the synergistic mechanism of SA with PMB, more experiments are needed. In terms of L-Ara4N-related resistance, a PMB-resistant strain needs to be constructed. Genes encoding LPS-modifying enzymes (*pmrE* gene) and regulators of the PmrAB/PhoPQ two-component systems (*mgrB* gene) will be focused on [69].

## 5. Conclusions

In conclusion, we found a natural product, SA, that could synergistically enhance the anti-GNB activity of PMB. SA was obtained by screening the natural product library and then was proven to promote the antibacterial activity of PMB in vitro by performing a checkerboard assay and constructing time–kill curves. In the in vivo peritonitis model, the antibacterial activity of PMB was enhanced by SA, with the survival rate, bacterial loads, and inflammatory responses significantly improved. Further, the detection of intracellular ATP levels, membrane potential and integrity measurements, electron microscopy observations, and metabolomic analysis indicated that the mechanism for the synergistic effect of SA with PMB occurred via membrane breakage enhancement. For the first time, SA has been shown to improve the antibacterial activity of PMB by damaging the bacterial membrane, assisting in reducing the dosage of PMB and constraining selective pressure. The synergy of SA with PMB also provides a promising approach to revive polymyxins for the treatment of GNB infections. Moreover, this research highlights further insights into combination strategies in drug development for infectious diseases.

## Figures and Tables

**Figure 1 pharmaceutics-16-00070-f001:**
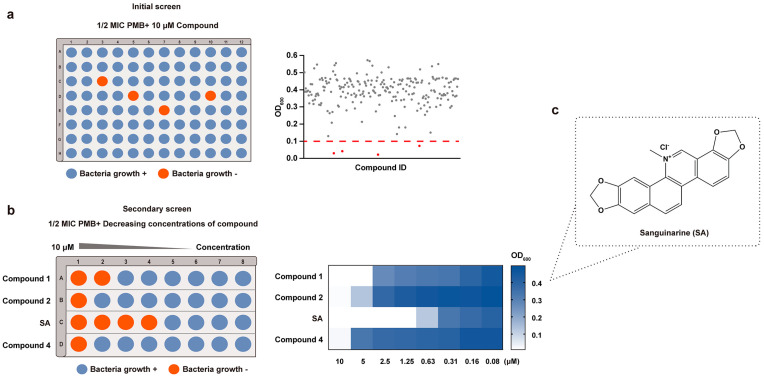
Identification of compounds that potentiate the antibacterial activity of PMB. (**a**) Left: diagrammatic drawing of the initial screen, which was conducted at 1/2 × MIC PMB in combination with 10 μM compounds against *E. coli* ATCC 25922; Right: a dot plot diagram of results that represents the screening data for the compounds, with the red dashed line delineating the cut-off for compounds with OD_600_ < 0.1. The four red dots represent the positive compounds in the initial screen. (**b**) Left: diagrammatic drawing of the secondary screen, which was conducted at 1/2 × MIC PMB in combination with the initial positive compounds at different concentrations starting from 10 μM in 2-fold dilutions; Right: OD_600_ of different combined treatments in the secondary screen. (**c**) Chemical structure of sanguinarine.

**Figure 2 pharmaceutics-16-00070-f002:**
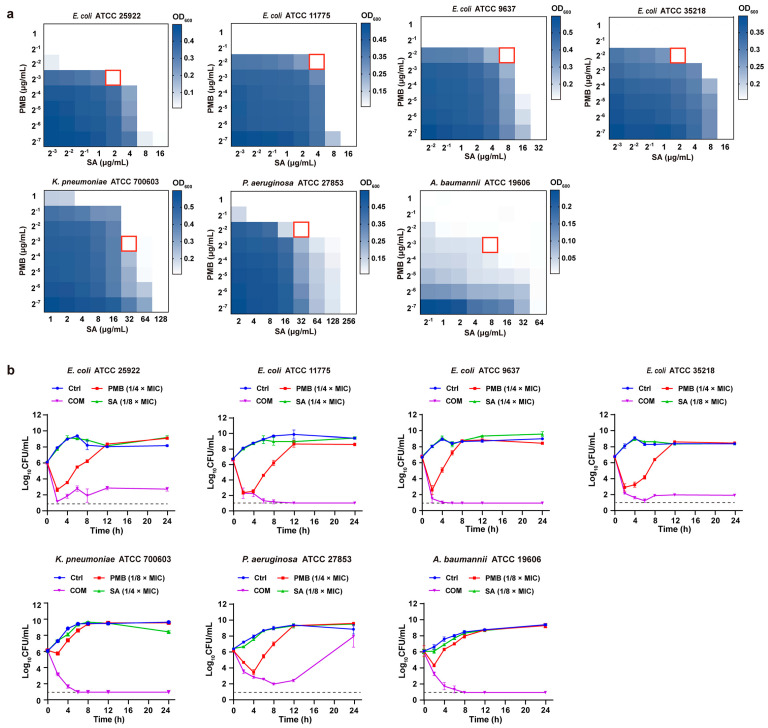
In vitro antibacterial activity of PMB-SA combinations against GNB strains. (**a**) Checkerboard assays of PMB-SA combinations. OD_600_ values were measured using a plate reader and transformed into a color gradient indicated by a scale: dark blue represents growth, while white represents no growth. The red boxes indicate combinations with the highest synergistic activity. (**b**) Time–kill curves of GNBs treated with PMB, SA, and PMB-SA combination with concentrations in FICI. The detection limit is indicated by a dotted line (10 CFU/mL).

**Figure 3 pharmaceutics-16-00070-f003:**
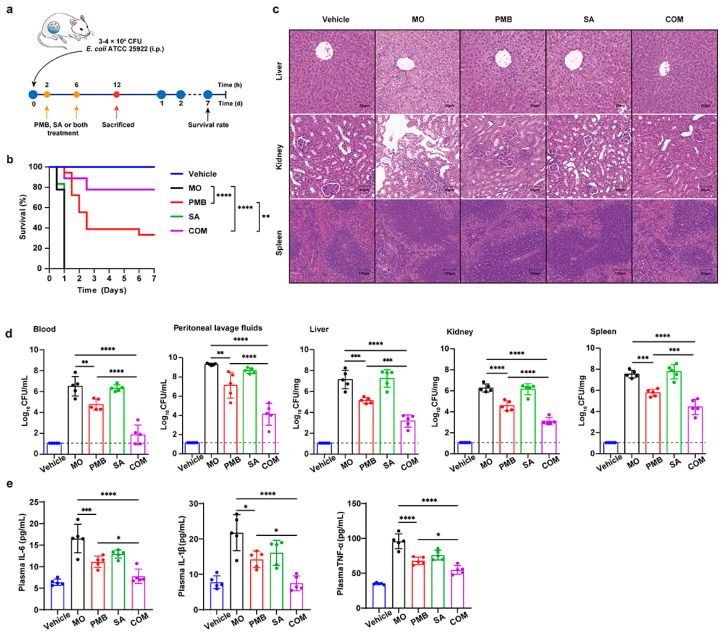
In vivo antibacterial activity of PMB-SA combination in murine peritonitis model. (**a**) Scheme of the experimental protocol. *E. coli* ATCC 25922 was intraperitoneally injected, and 0.5 mg/kg PMB and 10 mg/kg SA were used for treatment. (**b**) Animal survival rates within 7 days in each group (n = 18). (**c**) Pathological sections with H&E staining of livers, kidneys, and spleens in each group. Scale bars are 50 μm for livers/kidneys and 100 μm for spleens. (**d**) Bacterial loads were determined in blood, peritoneal lavage fluids, livers, kidneys, and spleens in CFU per ml and mg, respectively (n = 5). The detection limit is indicated by a dotted line (10 CFU/mL). (**e**) Assays of proinflammatory cytokines IL-6, IL-1*β*, and TNF-*α* in plasma according to ELISA (n = 5). All data are expressed as mean ± SD (error bars). * *p* < 0.05, ** *p* < 0.01, *** *p* < 0.001, **** *p* < 0.0001. Welch’s one-way ANOVA with Dunnett’s T3 multiple-comparison test was used. The survival rate was analyzed using a log-rank (Mantel–Cox) test.

**Figure 4 pharmaceutics-16-00070-f004:**
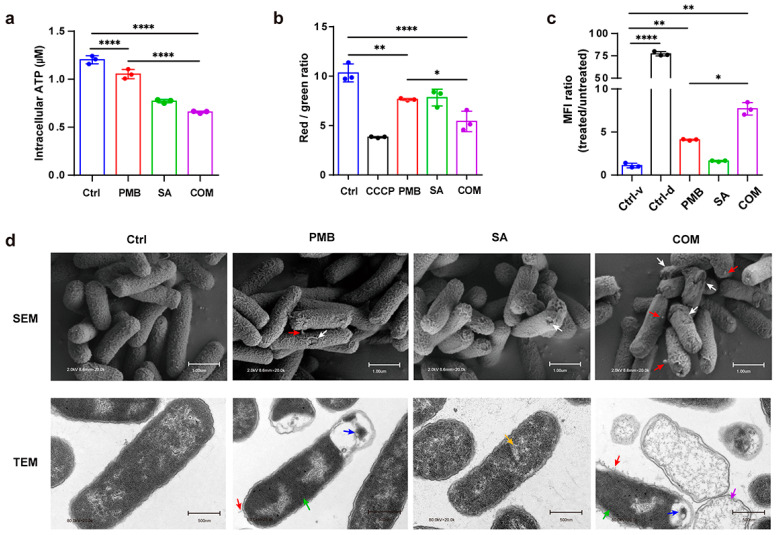
The synergy mechanism of SA with PMB. (**a**) Determination of intracellular ATP using a luminescence-based assay. (**b**) Fluctuation of bacterial membrane potential with the DiOC_2_ fluorescence-based method. (**c**) Changes in bacterial membrane permeability. The Ctrl-v group represents the viable control with untreated bacteria. The Ctrl-d group represents the dead control with bacteria killed by heating. MFI: median fluorescence intensity. The MFI ratio is defined as the MFI of each group over the MFI of the Ctrl-v group. (**d**) Morphological observations by scanning electron microscopy (SEM) and transmission electron microscopy (TEM). PMB (1/16 × MIC) and SA (1/4 × MIC) were used. Red arrow: tubular appendages; white arrow: damaged bacterial membrane; blue arrow: mesosomes; green arrow: inhomogeneous cytoplasm with abnormally increasing electron density; yellow arrow: loose cytoplasm; purple arrow: incomplete cell envelope structure. Scale bars were 1 μm and 500 nm, as indicated for SEM and TEM, respectively. All data are based on values obtained from three replicates (n = 3) and are expressed as mean ± SD (error bars). At least three technical replicates were run with each biological replicate. * *p* < 0.05, ** *p* < 0.01, **** *p* < 0.0001. Welch’s one-way ANOVA with Dunnett’s T3 multiple-comparison test was used.

**Figure 5 pharmaceutics-16-00070-f005:**
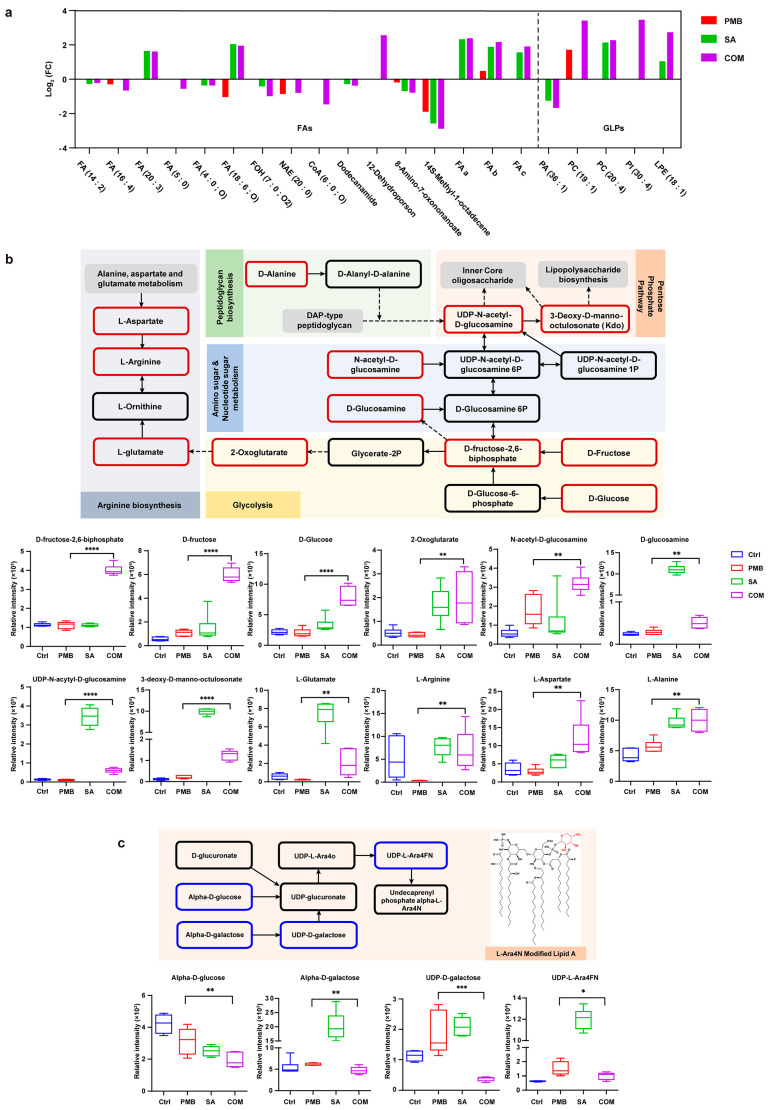
Metabolomic analysis associated with membrane perturbations of *E. coli* treated with PMB and SA. (**a**) Significantly perturbed fatty acids and glycerol phospholipids in *E. coli* ATCC 25922 treated with the compounds are shown in the bar graph (*p* ≤ 0.05). FAs: fatty acids; GLPs: glycerol phospholipids; FA_a_: Blumenol-CO-[rhamnosyl-(1->6)-glucoside]; FA_b_: 3-O-(*α*-L-arabinopyranosyl-(1->6)-*β*-D-glucopyranosyl) butyl 3R-hydroxybutanoate; FA_c_: Butyl (S)-3-hydroxybutyrate [arabinosyl-(1->6)-glucoside]. (**b**) Integrated pathway map of metabolites of *E. coli* ATCC 25922 significantly impacted by PMB, SA, and their combination in interrelated pathways: arginine pathway, glycolysis pathway, amino sugar/nucleotide sugar metabolism, and pentose phosphate pathway. Bar charts for the significantly impacted metabolites of these pathways following treatment with PMB (2 × MIC), SA (2 × MIC), or their combination for 1 h (*p* < 0.05). (**c**) Integrated pathway map of metabolites in *E. coli* ATCC 25922 significantly impacted by PMB, SA, and their combination in pathways of L-Ara4N-related PMB resistance. Bar charts for the significantly impacted metabolites of these pathways following treatment with PMB (2 × MIC), SA (2 × MIC), or their combination for 1 h (*p* < 0.05). Metabolites in red boxes were significantly increased after PMB-SA combination treatment compared to PMB alone, while in blue boxes were significantly decreased and in black boxes were no changes. Dotted arrows indicated indirect relationships between metabolites. * *p* < 0.05, ** *p* < 0.01, *** *p* < 0.001, **** *p* < 0.0001. All data were obtained from six replicates (n = 6) and are expressed as mean ± SD (error bars). Student’s *t*-test was used to analyze the results.

**Table 1 pharmaceutics-16-00070-t001:** MICs and FICIs of PMB with SA against Gram-negative bacterial strains.

Bacterial Strains	PMB (μg/mL)		SA (μg/mL)		FICIs
	MIC in Single Use	MIC in Combination	MIC in Single Use	MIC in Combination	
*E. coli* ATCC 25922	0.5	0.125	16	2	0.375
*E. coli* ATCC 11775	1	0.25	16	4	0.5
*E. coli* ATCC 9637	1	0.25	32	8	0.5
*E. coli* ATCC 35218	1	0.25	16	2	0.375
*K. pneumoniae* ATCC 700603	2	0.25	128	32	0.375
*P. aeruginosa* ATCC 27853	1	0.25	256	32	0.375
*A. baumannii* ATCC 19606	1	0.125	64	8	0.25

## Data Availability

The data used in the current study will be available from the corresponding author upon reasonable request.

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
