# Peer review of "Synergistic Activity and Mechanism of Sanguinarine with Polymyxin B against Gram-Negative Bacterial Infections"

_pharmaceutics, 2024, doi:10.3390/pharmaceutics16010070_

Round 1

Reviewer 1 Report

Comments and Suggestions for Authors

In the present manuscript, the authors identified Sanguinarine chloride to potentiate the antimicrobial activity of Polymyxin B in GNB. They studied the synergism in vitro and in vivo and finally through metabolomics, showed that SA-PMB synergism inhibits phosphorylation, nucleic acid biosynthesis and L-Ara4. The manuscript is clear and well-written and includes a variety of in vitro and in vivo studies. The metabolomics of the study is novel and adds a huge impact to the study. However, improvements to certain studies will be necessary to support the study’s central conclusion.

In vitro study:

·        Polymyxin B is antibiotic of last resort that is used during MDR infections. In the study, the authors studied the synergism is 7 strains, none of which are MDR bacteria. Observing the synergism is MDR strains should validate the conclusion.

·        All the 7 strains have PMB-MIC 2, which according to CLSI guidelines is in sensitive range. Will the SA-PMB synergism exist in PMB resistant strains?

·        A variety of mechanisms are proposed to mediate PMB synergism, including OMP OprH overexpression, efflux pumps and especially LPS modifications with 4-amino-4-deoxy-L-Ara. Discussion on the significance of SA-PMB synergism in the context of these resistance mechanisms is necessary.

In vivo study:

Histopathological observations are not so clear. For instance, ‘cells of livers and kidneys in the

model group were swollen’ is ambiguous. “White marrows of spleens in the model group were increased” should be supported by scoring.

In the methods section, the authors mentioned “At 12 h, the livers, kidneys, and spleens were aseptically removed for histopathology”. Is this 12 h post-infection or post-treatment? Either way, the number of CFU used for infection and extent of infection to these levels in the tissues in 12 h indicates E. coli strain to be extremely infective. Comment on the infection model. When was the blood CFU counts, RT-qPCR and ELISAs conducted?

Details of primers and antibodies used for RT-qPCR and ELISAs, respectively were not provided. How were the relative mRNA levels calculated?

Miscellaneous:

Sanguinarine prevents bacterial cell division in E. coli by inhibiting cytokinetic Z ring formation (PMID: 16342949). Is this observed in the present study (electron microscopy)?

Sanguinarine suppresses the 2-ketogluconate pathway of glucose utilization in Pseudomonas (PMID: 34566945).  Is this observed in E, coli (Metabolomics)?

Molecule 7 in PMID: 32562384 is Sanguinarine which was previously shown to synergize with colistin (polymyxin E). This should be cited in this work.

Statistics: I do not see any data where Students-t-test can be used. Please provide the details of stats in each figure legend. What was the post-hoc test used with one-way ANOVA?

Comments on the Quality of English Language

Minor edits are required for English language. 

Author Response

Reviewer 1

Comments and Suggestions for Authors

In the present manuscript, the authors identified Sanguinarine chloride to potentiate the antimicrobial activity of Polymyxin B in GNB. They studied the synergism in vitro and in vivo and finally through metabolomics, showed that SA-PMB synergism inhibits phosphorylation, nucleic acid biosynthesis and L-Ara4. The manuscript is clear and well-written and includes a variety of in vitro and in vivo studies. The metabolomics of the study is novel and adds a huge impact to the study. However, improvements to certain studies will be necessary to support the study’s central conclusion.

>We appreciate your positive comments for our work. We strongly agree with you that the current work needs further improvement. According to your constructive suggestions, we have made the following replies and meanwhile revised our manuscript.

In vitro study:

Polymyxin B is antibiotic of last resort that is used during MDR infections. In the study, the authors studied the synergism is 7 strains, none of which are MDR bacteria. Observing the synergism is MDR strains should validate the conclusion.

>Thanks for your suggestions. We are sorry we didn't make it clear that E. coli ATCC 35218 and K. pneumoniae ATCC 700603 are MDR strains available in our lab, both of which produce extended spectrum β-lactamases and are resistant to β-lactam antibiotics. Results in both Table 1 and Figure 2 showed the synergistic effect of PMB-SA in these two strains.

All the 7 strains have PMB-MIC ≤2, which according to CLSI guidelines is in sensitive range. Will the SA-PMB synergism exist in PMB resistant strains?

>Thank you for your suggestions. In our current results from metabolomics, the PMB-SA combination could block the L-Ara4N-related PMB resistance pathway, indicating a potential existence of PMB-SA synergism in PMB resistant strains. However, PMB resistant strains, which were hard to obtain, were needed to confirm this. We are now cooperating with qualified hospitals, trying to get the PMB resistant clinical isolates, and will carry out the antibacterial activity analysis of PMB-SA on these resistant strains in the future. These were stated in Line 609-615.

A variety of mechanisms are proposed to mediate PMB synergism, including OMP OprH overexpression, efflux pumps and especially LPS modifications with 4-amino-4-deoxy-L-Ara. Discussion on the significance of SA-PMB synergism in the context of these resistance mechanisms is necessary.

>Thanks for your suggestions. We strongly agree with you that the significance of PMB-SA synergism should be discussed. A variety of mechanisms are proposed to mediate PMB synergism, including the membrane perturbation, LPS modifications, efflux pumps and overexpression of the OMP OprH (PMID: 35595766, PMID: 35115797, PMID:25505462). Through metabolomics analysis, we found that SA enhanced membrane perturbations of PMB, consistent with the pharmacological results of synergistic activity for PMB-SA in membrane disruption. Meanwhile, after treating with PMB-SA, the changes in metabolites of L-Ara4N biosynthesis pathways, which have been proved to be associated with LPS modifications, were found. All these served as the hints for the further study on the significance of PMB-SA synergism. We have added these on the Discussion section in Line 604-615.

In vivo study:

Histopathological observations are not so clear. For instance, ‘cells of livers and kidneys in the model group were swollen’ is ambiguous. “White marrows of spleens in the model group were increased” should be supported by scoring.

>Thanks for your suggestions. We have made a more detailed description for the Results in histopathological observations. See Line 339-344. Meanwhile, the fold changes of white pulp area in spleens were scored statistically by Image Pro Plus in new Figure S1a.

In the methods section, the authors mentioned “At 12 h, the livers, kidneys, and spleens were aseptically removed for histopathology”. Is this 12 h post-infection or post-treatment? Either way, the number of CFU used for infection and extent of infection to these levels in the tissues in 12 h indicates E. coli strain to be extremely infective.

>Thanks for your suggestions. The time points "12 h" means 12 h post-infection. We have added this information in the revised version at Line 155. In our experiment, E. coli ATCC 25922 (3-4×104 CFU/mL) was used to induce murine peritonitis. The extent of infection in the model were entirely consistent with the previous results (PMID: 15891333, PMID: 14638760, PMID: 14500466, PMID: 11342656).

Comment on the infection model. When was the blood CFU counts, RT-qPCR and ELISAs conducted?

>Thanks for your suggestions. The blood CFU counts, RT-qPCR and ELISAs were conducted at 12 h post-infection. We have added this information in the revised version at Line 159.

Details of primers and antibodies used for RT-qPCR and ELISAs, respectively were not provided. How were the relative mRNA levels calculated?

>Thank you for your suggestions. The details of primers used for RT-qPCR were provided in supplementary materials (Table S1). The ELISAs were performed according to the instructions of mouse IL-6/IL-1β/TNF-α ELISA kit, purchased from Shanghai Enzyme-linked Biotechnology (China). This information was added at Line 95-96. The relative mRNA levels were calculated based on the 2^(-ΔΔCt) method according to Livak and Schmittgen (PMID: 11846609), which was added at Line 165.

Miscellaneous:

Sanguinarine prevents bacterial cell division in E. coli by inhibiting cytokinetic Z ring formation (PMID: 16342949). Is this observed in the present study (electron microscopy)?

>Thank you for your suggestions. Scientists in the research you mentioned (PMID: 16342949) expressed and purified recombinant FtsZ proteins in vitro and then observed changes in FtsZ proteins bundling (Z ring formation) after SA treatment by electron microscopy. The bacterial morphology was also examined. However, we focused on the surface and ultrastructural changes of E. coli treated by SA in our work. On this condition, the formation of Z ring and the disturbed cell division have not been observed, which may be related to the different treatment concentration and time of SA. However, the research you provided has important enlightening significance for further study on the antibacterial mechanism of SA, which we have cited in the revised version at Line 77-78.

Sanguinarine suppresses the 2-ketogluconate pathway of glucose utilization in Pseudomonas (PMID: 34566945). Is this observed in E. coli (Metabolomics)?

>Thanks for your suggestions. The periplasmic gluconate shunt (PGS) pathway that generates gluconate and/or 2-ketogluconate from glucose plays an important role in Pseudomonas species, which has been utilized in chemical production and medicine (PMID: 37454792, PMID: 16342949). However, in E. coli, the Embden-Meyerhof-Parnas (EMP) pathway has been reported to be predominant in glucose utilization (PMID: 22596202, PMID: 9811658). In this study, we have not observed the significant changes in 2-ketogluconate pathway from the results of metabolomics, which may be related to the metabolic characteristics of E. coli. Thank you for providing such important research in elucidating the multiple anti-infection mechanisms of SA. We have cited it in Line 79-80.

Molecule 7 in PMID: 32562384 is Sanguinarine which was previously shown to synergize with colistin (polymyxin E). This should be cited in this work.

>As you suggested, we have cited this work at Reference [57] in Line 572-573.

Statistics: I do not see any data where Students-t-test can be used. Please provide the details of stats in each figure legend. What was the post-hoc test used with one-way ANOVA?

>Thanks for your suggestions. The significance differences of some data were compared between two of the five experimental groups using Students-t-test, not all five groups together. We have made this clearly described in each figure legend of the revised version. Besides, the post-hoc test used with Welch’s one-way ANOVA was Dunnett’s T3 multiple comparison test, which was also added in Line 247.

Reviewer 2 Report

Comments and Suggestions for Authors

General: Heavy use of abbreviations which makes readability difficult. I would suggest reducing the overall number used in the manuscript.

Results: 

2.4 Natural products screening: which commercial library was used, were obtained from?

2.7 E. coli-induced peritonitits model

line148: which solvent was used? A better term here might be vehicle control.

2.8 Intracellular ATP levels & 2.10 Membrane Integrity

How many technical replicates were done with each biological replicate?

2.12 Metabolomics analysis 

line: 225. cells were ultrasound. Better term might be sonicated.

General comment on Figures:  Many of the graphs are very small and hard to read. In several of the figures it might be worthwhile to show the most pertinent graphs in the manuscript and place the others in supplemental material

line 364: font change

line 471: font change

lines 567 and 568. Staphylococcus should be with capital S

FICI: For cutoffs for FICI synergy, antagonism, indifferent/additive should have reference Odds, F.C (2003) Synergy, antagonism, and what the chequerboard puts between them. J Antimicrob Chemother doi: 10.1093/jac/dkg301

Comments on the Quality of English Language

Overall quality of english language is fine. 

Author Response

Reviewer 2

Comments and Suggestions for Authors

General: Heavy use of abbreviations which makes readability difficult. I would suggest reducing the overall number used in the manuscript.

>Thank you for your affirmation of our work. As you suggested, we have annotated all the abbreviations in the revised version and reduced the overall number used in the manuscript.

Results: 

2.4 Natural products screening: which commercial library was used, were obtained from?

>Thank you for your suggestions. The natural products commercial library, which was purchased from MedChamExpress (USA), was used in this research. We have added this information in the revised version at Line 93-94.

2.7 E. coli-induced peritonitits model

line148: which solvent was used? A better term here might be vehicle control.

>In the E. coli induced peritonitits model, 5% mucin was used as solvent. As you suggested, we have changed the term “sham” with “vehicle control” in the revised version.

2.8 Intracellular ATP levels & 2.10 Membrane Integrity

How many technical replicates were done with each biological replicate?

>Thank you for the suggestions. At least three technical replicates were done with each biological replicate. We have added this information in Line 181-182.

2.12 Metabolomics analysis 

line: 225. cells were ultrasound. Better term might be sonicated.

>Thank you for your careful reading. we have changed the term “ultrasound” with “sonicated” in the revised version at Line 232.

General comment on Figures:  Many of the graphs are very small and hard to read. In several of the figures it might be worthwhile to show the most pertinent graphs in the manuscript and place the others in supplemental material

>Thank you for your constructive suggestions. We have reorganized Figure 1, 3, 4 and Figure S1, S2 to make it easy to read.

line 364: font change

>Changed. See Line 371.

line 471: font change

>Changed. See Line 477.

lines 567 and 568. Staphylococcus should be with capital S

>Changed. See Line 564 and 565.

FICI: For cutoffs for FICI synergy, antagonism, indifferent/additive should have reference Odds, F.C (2003) Synergy, antagonism, and what the chequerboard puts between them. J Antimicrob Chemother doi: 10.1093/jac/dkg301

>As you suggested, we have cited this reference at Line 133.

Reviewer 3 Report

Comments and Suggestions for Authors

This manuscript describes synergistic effect of Sanguinarine with Polymyxin B against Gram-Negative bacteria in vitro and in vivo with a detailed explained mechanism of action. The only drawback of this study implies an absence of clinical isolates and isolates resistant to polymyxin B. In my opinion, some of these isolates should be introduced in this study so that the significance of what was described could be considered relevant for clinical application. This implies only checkerboard and time-kill assays.

Minor corrections are listed below:

Line 66-In clinics

Line 75-is an urgent

Line 77-87-This paragraph should be shorter, not like in abstract

Line 101- Big P change to small one im pneumoniae

Line 101-resistant to which antibiotic?

Section 2.4 (line 112)-Indicate which bacteria

Line 129- indifferent effect

Line 147- 2 and 6

Line 157-transcripton

Line 219- Overnight culture is in stationary phase not logarithmic

Line 222-32 mg/L is redundant

Line 246-Is not necessary to repeat E.coli

Figure 1. Secondary screen should be Fig1C, so please reorganize this Figure in accordance with text

Table 1 should be smaller

Figure 2A-is better to show colours with scale

                 -indicate in text that only P. aeruginosa grows after 24h

Line 307- assay is redundant word

Figure 3-resolution should be better

Figure 3A- correct to sacrificed

                 -it is better not to write PMB+SA, because in this experiment included different groups, it is better to write agents or something like that

Figure 3B-in this part should be reversed PMB and combination, because it is not in accordance with text and observations

Figure 3D-periotoneal

Line 364-change font for permeability

Figure 4C-In Figure legend explain better the image because I and II are not indicated in Figure;

                -Also, indicate the meaning of abbreviations ctrl-v and d

Line 492-change & to and

Discussion-the beginning part should be shorter, it is recommended not repeat parts from Introduction section

Lines 567, 568- Staphylococcus

Lines multiple-harmonize et al.

Line 571-573, 576-577, 607-610-Rephrase these sentences

Comments on the Quality of English Language

English in some sections like Discussion should be improved.

Author Response

Reviewer 3

Comments and Suggestions for Authors

This manuscript describes synergistic effect of Sanguinarine with Polymyxin B against Gram-Negative bacteria in vitro and in vivo with a detailed explained mechanism of action. The only drawback of this study implies an absence of clinical isolates and isolates resistant to polymyxin B. In my opinion, some of these isolates should be introduced in this study so that the significance of what was described could be considered relevant for clinical application. This implies only checkerboard and time-kill assays.

>Thanks for your constructive suggestions. We strongly agreed with you that PMB resistant isolates should be introduced to enrich the work with clinical significance. However, PMB resistant strains were hard to obtain. We are now cooperating with qualified hospitals, trying to get the PMB resistant clinical isolates, and will carry out the antibacterial activity analysis of PMB-SA on these resistant strains in the future.

Minor corrections are listed below:

Line 66-In clinics

>Changed. See Line 65.

Line 75-is an urgent

>Changed. See Line 74.

Line 77-87-This paragraph should be shorter, not like in abstract

>Thanks for your suggestions. We have revised the paragraph to make it shorter. The revised section was presented as below.

"Here, we reported a promising combination of SA with PMB for the treatment of GNB infection. We found that SA could promote the antibacterial activity of PMB in vitro and in in vivo murine peritonitis model, probably by enhancing the membrane breakage of GNB. SA was shown to be effective in potentiating the activity of PMB against a variety of GNB strains for the first time, which also highlighted the significant clinical potential for the treatment of GNB infections." See Line 84-89.

Line 101- Big P change to small one in pneumoniae

>Changed. See Line 104.

Line 101-resistant to which antibiotic?

>Thanks for your suggestions. The E. coli ATCC 35218 is an extended spectrum β-lactamases-producing strain and is resistant to β-lactam antibiotics. We have added this information in the revised version at Line 103-104.

Section 2.4 (line 112)-Indicate which bacteria

>Thanks for your suggestions. Bacteria used in this section was E. coli ATCC 25922, which have been added at Line 118.

Line 129- indifferent effect

>Changed. See Line 133.

Line 147- 2 and 6

>Changed. See Line 152.

Line 157-transcripton

>Changed. See Line 162.

Line 219- Overnight culture is in stationary phase not logarithmic

>Thanks for your careful reading. We are sorry to make such a mistake. We have deleted "overnight" in the revised version at Line 226.

Line 222-32 mg/L is redundant

>Deleted. See Line 228.

Line 246-Is not necessary to repeat E. coli

>Deleted. See Line 253.

Figure 1. Secondary screen should be Fig1C, so please reorganize this Figure in accordance with text

>Thank you for your suggestions. We have reorganized Figure 1 in accordance with the text, which made it easy to understand.

Table 1 should be smaller

>Thanks for your suggestions. We have changed the size of Table 1.

Figure 2A-is better to show colours with scale-indicate in text that only P. aeruginosa grows after 24h

>Thanks for your suggestions. Colour scales have been added in Figure 2A. Also, we have indicated in the text that only P. aeruginosa grows after 24 h in the time-kill assays. See Line 309-310.

Line 307- assay is redundant word

>Changed. See Line 313.

Figure 3-resolution should be better

>Thanks for your suggestions. We have reorganized Figure 3 with higher resolution.

Figure 3A- correct to sacrificed -it is better not to write PMB+SA, because in this experiment included different groups, it is better to write agents or something like that

>Thanks for your careful reading. As you suggested, we have corrected the word "sacrificed" and marked properly for different groups in Figure 3A. See new Figure 3A.

Figure 3B-in this part should be reversed PMB and combination, because it is not in accordance with text and observations

>We are sorry that we didn’t interpret it clearly. The survival rate for each group was calculated to evaluate the curative effect of PMB, SA and their combination. The results showed that all mice from the model group without treatment died within 24 h. When PMB was administrated alone, the survival rate was improved to 33.3%. Interestingly, the combination of PMB and SA significantly increased the survival rate to 77.8% (P < 0.01), more than 2.3 times fold than that of the PMB alone. These were stated in Line 321-325.

Figure 3D-periotoneal

>Changed. See new figure 3d.

Line 364-change font for permeability

>Changed. See Line 371.

Figure 4C-In Figure legend explain better the image because I and II are not indicated in Figure; -Also, indicate the meaning of abbreviations ctrl-v and d

>Thank you for your suggestions. The Ctrl-v group represented the viable control with untreated bacteria. The quadrant I was designated based on the dot-plot profiles of Ctrl-v. The Ctrl-d group represented the dead control with bacteria killed by heating. The quadrant II was designated based on the dot-plot profiles of Ctrl-d. We have added these explanations in the figure legend of Figure S2.

Line 492-change & to and

>Changed. See Line 498.

Discussion-the beginning part should be shorter, it is recommended not repeat parts from Introduction section

>Thank you for your suggestions. We have revised the beginning part of Discussion. The revised section was presented as below.

"The rapidly emerging of polymyxin-resistant strains brought the utility of polymyxin monotherapy into question. One of the strategies for overcoming polymyxin-relevant drug-resistance without augmenting toxicity is to decrease polymyxin exposure by combination with other agents. In this study, we found a natural product SA, which exhibited a synergistic antibacterial effect with PMB mainly by enhancing the bacterial membrane damage, providing a new insight for treating GNB infections." See Line 529-534.

Lines 567, 568- Staphylococcus

>Changed. See Line 564 and 565.

Lines multiple-harmonize et al.

> Thank you for your careful reading. We have harmonized the format of et al. See Line 570.

Line 571-573, 576-577, 607-610-Rephrase these sentences

>Thank you for your suggestions. We have rephrased these sentences as below.

"Since PMB is currently one of the most effective drugs available for severe GNB infections, SA which promote the activity of PMB has clinically significance." See Line 573-575.

"It has been reported that SA itself could reduce the inflammatory response of macro-phage in LPS-induced peritonitis [58], which might also contribute to the synergistic activity of PMB-SA in the E. coli-induced peritonitis model." See Line 575-578.

"Genes encoding LPS-modifying enzymes (pmrE gene), and regulators of the PmrAB/PhoPQ two-component systems (mgrB gene) will be focused [68]." See Line 615-616.

Reviewer 4 Report

Comments and Suggestions for Authors

This is a very extensive study that investigates synergistic effect of antibiotic polymixyn B with a natural product sanguinarine. I have several minor comments to the authors:

Introduction:

- line 46 and 51 - in one of those sentences replace "besides" with another word.

- line 66 - instead "clinical" use "clinical setting" or something similar, a noun is missing

 - line 76 -  GNB or MDR-GNB or both? Since this paragraph refers to MDR-GNB

- lines 77-87 - this section is more suitable for discussion and/or conclusion, maybe more information might be given about sanguinarine and information regarding it's biological activity, possible targets, mechanism of action, since this hasn't been mentioned earlier in the text

- Results: figures 3-5 are to extensive, maybe the authors might divide them into more smaller ones, and divide them throughout the text, it might be more convenient for the readers. Also, all abbreviations in the figures should be included also in the text describing the figure (text below the figure

- Lines 394 and 488 - delete extra spacer

- "alpha", "beta" - in the names of the compounds - replace with the signs

Conclusion: please rewrite more detailed conclusion.

Author Response

Reviewer 4

Comments and Suggestions for Authors

This is a very extensive study that investigates synergistic effect of antibiotic polymixyn B with a natural product sanguinarine. I have several minor comments to the authors:

>We sincerely appreciate your positive comments for our work. According to your suggestions, we have answered all the questions and revised the manuscript.

Introduction:

line 46 and 51 - in one of those sentences replace "besides" with another word.

>Thanks for your suggestions. We have replaced "besides" with "in addition" in Line 50.

line 66 - instead "" use "clinical setting" or something similar, a noun is missing

>Thanks for your suggestions. We have corrected "clinical" to " clinics" in Line 65.

line 76 - GNB or MDR-GNB or both? Since this paragraph refers to MDR-GNB

>Thanks for your suggestion. Here it should be MDR-GNB. Sorry to make you confused and we have changed the word "GNB" to "MDR-GNB" in Line 75.

lines 77-87 - this section is more suitable for discussion and/or conclusion, maybe more information might be given about sanguinarine and information regarding its biological activity, possible targets, mechanism of action, since this hasn't been mentioned earlier in the text

>We agree. As you suggested, we have revised this section to make it shorter. Besides, we added more information on SA. The revised section was presented as below.

"Sanguinarine (SA) is a benzophenanthridine alkaloid extracted from Sanguinaria canadensis, showing anti-GNB activity through various mechanisms. SA restrained proliferation of GNB by blocking cytokinesis through hindering FtsZ assembly and bundling or inhibited P. aeruginosa 2-ketogluconate utilization by interfering with the function of KguD or KguK proteins [20,21]. Besides, SA effectively synergized with aminoglycoside in killing on GNBs due to the enhanced membrane permeability and increased intracellular ROS level [22]. Thus, SA has the potential to synergize with other antibiotics to increase the anti-bacterial effect.

Here, we reported a promising combination of SA with PMB for the treatment of GNB infection. We found that SA could promote the antibacterial activity of PMB in vitro and in in vivo murine peritonitis model, probably by enhancing the membrane breakage of GNB. SA was shown to be effective in potentiating the activity of PMB against a variety of GNB strains for the first time, which also highlighted the significant clinical potential for the treatment of GNB infections." See Line 76-89.

Results: figures 3-5 are to extensive, maybe the authors might divide them into more smaller ones, and divide them throughout the text, it might be more convenient for the readers. Also, all abbreviations in the figures should be included also in the text describing the figure (text below the figure)

>Thanks for your suggestions. We have reorganized Figure 1,3, 4 and Figure S1, S2 to make it easy to read. In addition, we have described all the abbreviations of the manuscript in "Abbreviations" section. See Line 632-640.

Lines 394 and 488 - delete extra spacer

>Changed. See Line 403 and 494.

"alpha", "beta" - in the names of the compounds - replace with the signs

>Changed. See Line 461, 462, 491.

Conclusion: please rewrite more detailed conclusion.

>Thanks for your suggestions, we have rewrite more detailed conclusion in the revised version. See Line 618-631.

Reviewer 5 Report

Comments and Suggestions for Authors

Multidrug-resistant Gram-negative bacteria (MDR-GNB) are now an important problem in infection control. The authors evaluated the synergistic activity of the combination use of sanguinarine and polymyxin B against MDR-GNB. They performed this study both in vitro and in vivo, which can be appreciated.

This study is well designed, and the results of this study are reliable. This study will contribute to the control of MDR-GNB infections. 

However, I have a concern about the safety of the combination use.

Polymyxin B has a strong adverse effect on renal function. Sanguinarine has a cytocidal effect on animal cells.

The authors should discuss the risk that the combination use of sanguinarine and polymyxin B leads to severe adverse effect.

Author Response

Reviewer 5

Comments and Suggestions for Authors

Multidrug-resistant Gram-negative bacteria (MDR-GNB) are now an important problem in infection control. The authors evaluated the synergistic activity of the combination use of sanguinarine and polymyxin B against MDR-GNB. They performed this study both in vitro and in vivo, which can be appreciated.

This study is well designed, and the results of this study are reliable. This study will contribute to the control of MDR-GNB infections. 

However, I have a concern about the safety of the combination use.

Polymyxin B has a strong adverse effect on renal function. Sanguinarine has a cytocidal effect on animal cells. The authors should discuss the risk that the combination use of sanguinarine and polymyxin B leads to severe adverse effect.

>We sincerely appreciate your kind time and positive comments for our works. We agree with your concern about the safety of PMB-SA combination use. For SA, cytotoxicity induced by binding to Na+-K+-ATPase has been reported to be important for limiting its clinical applications (PMID: 29616225). However, SA exhibited relatively low toxicity by oral administration or in ionic form (PMID: 20385218). In addition, Cao et al. found a promising anti-cancer lead compound with higher selective cytotoxicity through SA-based structural modification, indicating the potential use of SA in anti-infectious diseases through structural modification (PMID: 25444843). For polymyxins, colistin and PMB are two currently available types, both of which were considered to have nephrotoxicity. Recent studies have shown that the prevalence of nephrotoxicity was significantly higher in colistin than in PMB (PMID: 25266910, PMID: 27686609). Due to the dose-limiting adverse effect of nephrotoxicity, reducing the daily doses of PMB lead to the decreasing risk of nephrotoxicity, which will subsequently result in subtherapeutic antibiotic concentrations and diminish the bacterial killing. However, our study showed that the dose of PMB was reduced when combined with SA without attenuated efficacy. Thus, combinations of SA with low-dose PMB might be promising in the clinical use. These were stated in the Discussion section at Line 579-592.